# Conceptual Challenges on the Road to the Multiverse

## Ana Alonso-Serrano [1,*,†] and Gil Jannes [2,†]

[1] Max Planck Institute for Gravitational Physics, Albert Einstein Institute, Am Mühlenberg 1, D-14476 Golm, Germany
[2] Department of Financial and Actuarial Economics & Statistics, Universidad Complutense de Madrid, Campus Somosaguas s/n, 28223 Pozuelo de Alarcón (Madrid), Spain; gil.jannes@ucm.es
* Correspondence: ana.alonso.serrano@aei.mpg.de
† The authors contributed equally to this work.

**Abstract:** The current debate about a possible change of paradigm from a single universe to a multiverse scenario could have deep implications on our view of cosmology and of science in general. These implications therefore deserve to be analyzed from a fundamental conceptual level. We briefly review the different multiverse ideas, both historically and within contemporary physics. We then discuss several positions within philosophy of science with regard to scientific progress, and apply these to the multiverse debate. Finally, we construct some key concepts for a physical multiverse scenario and discuss the challenges this scenario has to deal with in order to provide a solid, testable theory.

**Keywords:** multiverse; physical multiverse; philosophy of science; empirical testability; falsifiability; Bayesian analysis; fine-tuning

## 1. Introduction

Ideas related to what we presently call "the multiverse" have historically always attracted both supporters and detractors. The multiverse is not really a theory, but a scenario that arises in several theories and can be defined in different ways depending on the underlying theory. This apparently vague remark in fact has deep consequences on the discussion about the possible existence of other universes and the associated change of the very definition of "the universe", but also of what a scientific theory is or should be, and how it should be assessed. As we will discuss later in detail, there is no standard or commonly agreed upon definition of universe, and therefore not for the multiverse either. What seems clear along history is that the universe has been defined as the (connected) region of space accessible to us. However, this meaning has evolved depending upon the theory and observations at our disposal.

In the past few years, the amount of papers and ideas about the multiverse have increased tremendously. The sometimes heated discussion about the viability of the multiverse and related ideas [1–3] show that we are far from reaching an agreement in the scientific community. At this stage there are, on the one hand, strong claims about what "could be one of the most important revolutions in the history of cosmogonies" [4], which "changes the way we think about our place in the world" [5] and "if true - would force a profound change of our deep understanding of physics" [4]. On the other hand, there is also a strong opposition to the very mention of the possibility of a multiverse scenario and of the scientific significance of such mentions [6,7].

We therefore believe that it is relevant to examine the fundamental questions that emerge in multiverse ideas, and to emphasize some criteria that the multiverse should meet before being more widely acceptable as a viable scientific proposal. We therefore undertake an analysis based on concepts from the philosophy of science related to the definition of the scientific method, an epistemological

theory, and a scientific revolution implying a change of paradigm. This reflection will help us postulate some constraints that should be imposed on multiverse theories and the conceptual challenges they will need to confront.

## 2. The Multiverse: Nothing New under the Suns?

The idea that there might be other worlds or universes beyond our own has been a recurrent concept throughout history [8,9]. It followed naturally from the desire of knowing whether we are unique observers, and of the possibility of discovering worlds similar to ours. At the same time, it has always been accompanied by skepticism about the possibility of actually answering these questions.

These issues are part of the most ancient and most essential philosophical and cosmological questions. It should therefore come as no surprise that the first recorded notion of a multiverse in occidental intellectual history dates back to the ancient Greeks. Anaximander, in the 6th Century BC, speculated about a plurality of worlds such as our cosmos, appearing and disappearing in an eternal movement of generation and destruction. A few centuries later, Epicurus described how an unlimited number of worlds fills the infinite vacuum [10].

In medieval times, Robert Grosseteste described the condensation of different universes from an initial big-bang-like explosion [11]. Giordano Bruno proposed an infinite "cosmic pluralism" filled with many inhabitable worlds as an alternative to the Copernican heliocentric model [12]. In the 18th century, Emanuel Swedenborg conjectured a model of the evolution of our solar system and the firmament we observe, based on theological and philosophical arguments. He postulated the possible existence of other celestial spheres in the firmament and argued that every of those world-systems would follow the same principles. This argument can be interpreted as the first idea of a nebular hypothesis [13]. Thomas Wright was the first to interpret the astronomical observations of distant faint nebulous structures as other galaxies, suggesting that they could have their own "external creation" [14]. This idea was later elaborated by Immanuel Kant, who popularized the idea of the possible existence of habitable worlds around stars other than the Sun [15]. This theory was baptized "island universes" by von Humboldt a century later [16]. By 1920, as more observations became available, the question led to the "Great Debate" between Shapley and Curtis about the scale of the Universe. Shapley defended that our Milky Way constituted the entire universe and that the other observed nebulae were small entities on its outskirts, while in Curtis' opinion, at least some of them were in fact separate, distant galaxies [17]. It was not until Hubble's decisive observational evidence a few years later that this battle of paradigms, originally started from purely philosophical principles, was finally settled.

From a more metaphysical point of view, Leibniz argued that our universe was the best among an infinity of possible universes [18]. According to Leibniz, logical constraints meant that (even) God could not have made our universe any better. This argument was later turned around by Schopenhauer, who argued that our world must be the worst of all possible worlds, because if it were even slightly worse in any respect, life could not continue to exist [19]. This argument is curiously reminiscent of recent fine-tuning arguments: a slight change in any of several basic constants would have made complexity (and therefore life) impossible [20].

Before turning to contemporary theories of the multiverse, it might be interesting to mention that there also exist (non-occidental) religious and mythological ideas related to the multiverse. Such ideas are implicit in Buddhism, with its cyclical view on the continuous destruction and recreation of the universe. They appear explicitly in Hinduism: "The golden egg that is this universe is wrapped in seven sheaths: the earth and the other elements, each casing being ten times as great as the one it encases. There are millions upon millions of these in each universe. There are millions of such universes. Lord, all these together are like a single atom upon your head! So, we call you Ananta, Infinite One." [21].

In Christianity, the multiverse idea is controversial, as it seems opposite to the uniqueness idea common to most monotheistic religions. However, Page has defended that the multiverse is not in

conflict with Christianity [22]. In this context, Page mentions a serious challenge to the multiverse concept, namely the question of whether sinning civilizations in other universes have also been redeemed by the death of Christ. To our great relief, Page himself answers this question: "we could just interpret the Bible to mean that Christ's death here on earth is unique for our human civilization", and so with peace in mind we will focus here on cosmological approaches of the multiverse and its conceptual challenges.

## 3. Definition and Classification of the Multiverse

The epistemological extension from the universe to a multiverse is often compared to the Copernican revolution, as a further step in the gradual loss of importance of our own habitat (although there seems to be some disagreement about whether this would be the fourth [4,23] or fifth [24] Copernican revolution). However, from a physical point of view, the contemporary cosmological concept of multiverse arises not so much as a direct theory in itself, but as an indirect consequence of problems mainly related to the current cosmological paradigm of an accceleratedly expanding universe governed by the laws of General Relativity [1]. The multiverse is argued to be a natural extension of developments within string theory or early-universe cosmology (in particular, chaotic eternal inflation), and is invoked to solve a series of open problems in theoretical physics, such as the problem of the beginning of the universe, the cosmic coincidence problem, or the smallness of the cosmological constant, as well as the more general fine-tuning of physical constants [1]. In some of those cases, the multiverse in itself only partially solves the problem, but mainly establishes a reformulation of the question. For example, related to the fine-tuning problem, the multiverse suggests a distribution of values of certain fundamental constants among the different possible universes. The question why we live precisely in a universe with the observed values can then be answered by some form of the anthropic principle.[2] The new question which arises then is how likely the values of the physical constants of our universe are across the probability distribution within the multiverse. In this context, Vilenkin introduced the mediocrity principle [27], which defends that we should be "typical" observers, and therefore a priori we are expected to live in one of the most probable universes among all those which allow for the existence of life. Ideally, this principle will be testable by comparison with the probability distribution. We will come back to this issue later in Section 4.2.5. Let us first look at possible definitions of the multiverse.

In general terms, a first attempt to define the concept of multiverse could be that the multiverse encompasses all the multiple possible universes predicted by an underlying theory insofar as they are actually realized, i.e.: everything that physically exists, the totality of space and time and its material-energetic content. However, this (intentionally vague) definition leads to an obvious question from a semantic point of view. If we understand the Universe etymologically as the whole possible entity, all of spacetime and its content, then there is no place for the multiverse. Perhaps one should then redefine the universe itself, depending on concepts such as causal connection or variations of physical laws. The situation is reminiscent of the atom, originally meaning "indivisible". Eventually the physical meaning was overcome even though the name stuck. In the case of the multiverse, this apparently lexicological question bears a direct impact on physical issues. As is well-known, in the

---

[1]　Even though Everett's many-worlds interpretation of quantum mechanics is presently considered a multiverse scenario (see Tegmark's classification below), it really stands a bit apart for a variety of reasons, the first one being its historical origin purely within quantum mechanics. We will briefly mention this scenario again in Section 3.1 but otherwise focus mainly on cosmological multiverse scenarios.

[2]　For the relationship between the anthropic principle and the multiverse, see e.g., [25]. We will not discuss the anthropic principle here because, first, as paraphrased in [25], "many commentators have already thrown much darkness on this subject, and it is probable that, if they continue, we shall soon know nothing at all about it"; and second because, although the anthropic principle has undoubtedly contributed much to conceptual thinking about the multiverse, it is not clear whether it can also make any real contribution when it comes to empirical predictions, let alone—despite common claims to the contrary—whether it has done this so far [26].

case of a single universe, the boundary conditions are crucial to determine the mathematically and physically acceptable solutions (see the Hartle–Hawking no-boundary proposal [28] or Vilenkin's tunneling proposal [29]). Then in the case of a multiverse, the issue of the boundary conditions between the various universes within the multiverse is probably equally important [30]. In fact, the issue is even more general, since the overall nature of the multiverse depends on the particular definition one uses of the constituent universes. It is possible to consider as universe, for instance, the habitable region we live in (delimited by the Hubble sphere); a causal spacetime region; one of the quantum branches in the Everett interpretation of quantum mechanics; or simply one of the particular solutions to the cosmological equations that appear in string theory.

The best-known classification for the different multiverse hypotheses is due to Tegmark [31]. Tegmark establishes a hierarchical classification, where each higher level includes the lower ones. The level I multiverse consists of a variety of Hubble volumes, causally disconnected but all with the same physical laws and constants. Level II allows for a variation of the physical constants, for example due to bubble-breaking during the inflationary phase. Level III corresponds to quantum many-world branching. Finally, level IV is constituted by all the different possible mathematical structures, all of which are assumed to represent physically real universes.

A point which might be worth mentioning is that different physical multiverse models are not always straightforward to classify in Tegmark's (or some other) scheme. More generally, both when defending or criticizing the multiverse, or when trying to elaborate on "the multiverse", it is not always clearly stipulated which kind of multiverse one is dealing with, and this can seriously complicate the assessment of the arguments.

In the following, we focus on the origin of the most common contemporary multiverse models.

### 3.1. Physically Motivated Multiverse Scenarios

The earliest multiverse model in modern physics comes from Everett's many-worlds interpretation of quantum mechanics [32]. This interpretation considers the multiverse as all the possible histories from a quantum superposition, with one particular branch corresponding to our universe. This configuration of the multiverse as a host of bifurcating quantum branches leads to a continuous multiplication of parallel universes. Unsurprisingly, this interpretation has historically been quite controversial.

Inflation theory has led to a different multiverse notion. As a consequence of the quantum effects in the early universe, it could be possible to create new universes by a mechanism of eternal chaotic inflation [33–35], in which the different regions of space can transform into macroscopic bubble universes, which split off from a preceding universe and create a new one. In these multiverse models, the fundamental laws of physics are the same in each universe, due to the fact that they were originated in a common inflationary universe. The constants of nature, however, are allowed to have different values, depending on the specific inflationary process of each particular universe.

In the context of string theory, the idea of a multiverse stems from the pocket universes associated with the colossal number of false vacua predicted by the theory, which conform the so-called landscape [35,36]. Each of the universes could have different dimensions, elementary particles or fundamental constants of nature. In this scenario, our universe emerges by a selection procedure, following an anthropic reasoning [36], or arguments from quantum cosmology [37]. This approach can be related with the idea of bubble universes in the sense of the possibility of tunneling among different vacua, giving rise to an eternal inflation that populates the landscape.

Recently, the idea that the landscape should be constrained by consistency conditions has gained momentum. The argument is that the vast range of solutions coming from the landscape are physically restricted to those that give rise to effective field theories, surrounded by a swampland of inconsistent solutions [38]. In this scenario, the huge amount of possible vacua from the string landscape is strongly restricted, possibly leading to a unique vacuum state and hence without room for a string multiverse, except perhaps in the form of a cyclic universe [39]. This idea is closely related with other proposals

about cyclic universes, where each end of a universe poses the initial conditions for a next one, thus leading to a conformal cyclic cosmology [40].

There exists a larger variety of multiverse scenarios originating from other physical ideas and proposing different concrete schemes. We should stress that the different multiverse scenarios, although conceptually akin, are so far only vaguely related in terms of physical formulation. So an obvious challenge on the road to the multiverse is to clarify the physical and mathematical relation between the different multiverse scenarios. For example, the relation between inflation and string theory is a subject of ongoing research, and is in fact considered one of the major challenges within string theory research, see e.g., [41] and references therein. According to the present status of research, it seems that only certain very specific string theory scenarios might actually allow for inflation, and the concrete details of the mechanism are only starting to be understood quantitatively. To illustrate this point, ref. [41] states that "At present, we are led to inflation in string theory by a web of inference" and that a better understanding of "non-supersymmetric solutions of string theory, particularly de Sitter solutions (...) continues to be a zeroth-order challenge for deriving inflation from string theory". Curiously, the fact that this embryonic understanding of the relation between inflation and string theory poses a major challenge for the multiverse idea and especially for the interpretation that a common view should exist between string and inflationary multiverse scenarios is a question that (to the best of our knowledge) is barely being addressed in current research. With respect to Everett's many-worlds interpretation, the question is perhaps even less clear. In [42], an argument was made for a relation between the many-worlds interpretation and the (inflationary) multiverse. However, it is probably only fair to say that the argument is mainly qualitative and much further work is required in this area to show whether the conjectured connections between the different types of multiverse can actually be described in a concrete and convincing way.

A related point is that all of these multiverse scenarios currently face a series of unsolved issues and require much more detailing before they can be considered mature physical theories. This, in combination with the simple fact that the multiverse can be argued to constitute a change of paradigm, makes the multiverse subject to criticism. The strongest argument against the multiverse probably lies in the fact that the multiverse is considered a speculative idea that cannot be falsified, perhaps not even in principle. Indeed, one could wonder what physical sense it makes to consider other different universes if these have no detectable effect whatsoever on our universe, possibly not even in principle. Some scientists argue that, if this is indeed the case, then the multiverse cannot be considered a scientific theory, but should at most be included in the field of metaphysics [1]. Regardless of the defense that some authors have made of "non-empirical theory assessment" [43] and related concepts (see also our discussion in Section 4), it bears little doubt that multiverse scenarios would gain strength if they would lead to new and preferably concrete predictions about the properties of our universe [44]. Interesting examples are the prediction that some features of the CMB spectrum, such as the cold spot, could be due to entanglement with another universe in string theory [37,45,46], collisions with other bubble universes in the context of eternal inflation [47–49] or recently topological defect nucleation in bubble universes [50]. From a philosophical point of view, as stressed by Popper [51], it is better to have concrete conjectures that can be tested and possibly refuted, rather than a very general scenario which cannot be confirmed nor refuted.

Before discussing this philosophical question in more detail, let us already mention a lesser-known multiverse scenario, which could alleviate some of the mentioned criticisms. In the proposal of a quantum multiverse [52–55], a quantum cosmology program is developed for the multiverse as a set of entangled universes. Two relevant characteristics of this scenario are the following. First, it does not pre-suppose a specific model for the different universes to pop up. Second, the entanglement between universes could give rise to dynamical and observable effects on each universe due to its interaction with other universes [56–58]. We will come back to this idea in Section 6.

As we have just argued, a question that naturally emerges when dealing with such physical theories that explore the limits of our knowledge concerns their scientific viability. In this context, let

us take a look at the descriptions of scientific method, change of paradigm and related issues in the philosophy of science.

## 4. Philosophical Aspects

There are various reasons why it is relevant to look at philosophical aspects of the multiverse question. Let us just mention two. First, as we already mentioned in the introduction, the generic idea of a multiverse is not really new, and so it might be interesting to look at what philosophers and historians of science have to say about this.

Second, theories about the multiverse almost automatically lead to questions about how to assess these theories. Notions such as theory confirmation, verification and viability then become important. These notions are rarely defined explicitly, but have acquired relatively well-developed meanings in the philosophy of science. The following (basic) definitions might be useful to set the vocabulary. "Theory assessment" consists of submitting a given hypothesis to empirical data. As possible outcomes, "theory confirmation" consists of empirical evidence which supports a given hypothesis, in particular by being in agreement with a prediction from the hypothesis. "Falsification" obviously is the opposite case: empirical evidence which contradicts a given hypothesis. "Verification" is the ideal case in which supporting evidence is so strong that the hypothesis can be considered to be conclusively confirmed. Finally, the "viability" of a theory could be paraphrased as its compatibility with already existing data, irrespectively of whether it has produced any new predictions which could be submitted to additional testing. We will not further discuss these concepts explicitly, but the remainder of this section will make clear that their understanding has evolved over time, and that they are much more involved than the naive definitions just given might suggest, see e.g., [59].

Our third, and most important argument, is that the idea of a multiverse clearly challenges the epistemological boundaries of science, and so enters into the grey zone where physics meets philosophy. In fact, the multiverse is often presented as a change of paradigm which completely alters our understanding of cosmology, and perhaps even more: some physicists claim that the multiverse revolutionizes the way in which we should look at science itself, and the way in which we assess scientific theories. However, these terms, "paradigm" and "scientific revolution" stem from the philosophy of science, where they were studied in great detail. Claims about the character of science and its methodology transcend science itself, and would thus benefit from a broader philosophical context.

The philosophical dispute about the multiverse has so far taken place almost exclusively between physicists, with professional philosophers largely staying safely out of the ring. The argument has centered on (naive interpretations of) Popper's falsificationism criterion, with some recent attempts to reframe the question in terms of Bayesianism. In the light of the philosophy of science of the past century, this is a bit curious. To explain why, it might be useful to go through a crash course which will lead us from Popper to Bayesianism, and then see how these ideas can be interpreted in the context of the multiverse.

### 4.1. Philosophy of Science and the Description of Scientific Progress

We will limit ourselves to the key elements of Popper's, Kuhn's, Lakatos' and Feyerabend's ideas with relation to the demarcation problem and the formal description of scientific progress, and how Bayesianism can be situated in this context. The interested reader is referred to, e.g., [60] or [61] for good introductions.

Let us start by sketching the historical context in which Popper came to prominence. In the early 20th Century, there was a generalized expectation that all of mathematics could be framed on solid logical foundations. This expectation was epitomized by Russell and Whitehead's "Principia Mathematica" [62–64] and Hilbert's programme [65,66]. A related feeling existed in the philosophy of science: science should obey firm laws of logic, and scientific progress should be formally expressible in terms of logical laws related to deduction and induction. The main discussion in the philosophy

of science of the epoch was between proponents and opponents of logical positivism: the idea that (both in science and in philosophy) only verifiable claims about the empirically observable reality are meaningful. However, even the opponents (including Popper) of logical positivism opposed only its positivist part but had no doubt that scientific progress could be described as a cumulative logical process. The Russell–Whitehead–Hilbert ambition was blown to pieces by Gödel's incompleteness theorems [67]. The demise of logic as the foundation of scientific knowledge took a bit longer, starting with Kuhn. However, we are running ahead of our argument.

Karl Popper's landmark "Logik der Forschung" [68] was an attempt to circumvent the problem of induction while retaining the logical mould into which the description of scientific process should be cast. The problem of induction, in its simplest version, is the fact that inference from (no matter how many) concrete observations or experiments can never lead to certain knowledge about a general theory. Popper's replacement of induction as a scientific criterion by falsification, and by a process of conjectures and refutations, is logically water-tight: in principle, a single counter-example suffices to demonstrate the falsity of a general conjecture. However, Popper's programme failed in two senses. First, it failed as a demarcation criterion to distinguish science (which makes falsifiable conjectures) from non-science (which does not). Second, it failed as a description of how scientific progress really works in practice. Enter Thomas Kuhn.

Kuhn held a historical view of science, as opposed to Popper's normative view. In [69], Kuhn defended that the advance of scientific knowledge is not linear and continuous, but proceeds by alternation between normal science and paradigm shifts or scientific revolutions. The paradigm is the basic set of concepts, beliefs and practices shared by a community of scientists. During the normal science phase, scientists attempt to articulate this paradigm in more detail, make empirical interpretations and predictions, but do not question the paradigm itself. The observational interpretations are always framed within the paradigm, and in particular: the further they are distant from direct sensory experiences, the more they depend on theoretical concepts characteristic of the paradigm, an issue called the "theory-ladenness of observations".

Through accumulation of anomalies, i.e.: conceptual or observational challenges that resist easy solution within the paradigm, this paradigm can enter into a state of crisis, thus opening the possibility for a paradigm shift or scientific revolution. Some scientists will swap allegiance to a new paradigm, others will stick to the old paradigm until they retire (or die). However, generally speaking, scientists usually adhere quite strongly to the fundamental assumptions of their own paradigm, which are often not even formulated very explicitly, and use these to judge other paradigms. This lack of an explicit formulation of criteria within each paradigm makes it very hard for scientists belonging to different paradigms to converse rationally about the pros and cons of each approach, a problem called "incommensurability". This is related to the well-known problem of underdetermination of theory by evidence: the idea that the experimental and observational evidence available within a particular branch of science is (even in principle) insufficient to pick out a single "true" theory in a uniquely determined way.[3] Kuhn's response to the problem of underdetermination is that "scientific truth" is not solely determined by objective facts but also by consensus within the scientific community. In this sense, Kuhn was probably the first to emphasize the social aspect of science, an aspect which was later worked out in detail by authors such as Pickering [73]. This, in combination with the lack of a clear criterion for paradigm change, has led to criticisms on Kuhn as proposing a relativist, even irrational view on the progress of science. Kuhn defended himself by pointing out that rational criteria for paradigm choice can indeed be identified, such as empirical accuracy, consistency, broad scope, simplicity, and fruitfulness. However, these criteria are not sufficient, in Kuhn's view: two scientists might agree on the criteria but nevertheless make different paradigm choices.

---

[3] This issue becomes ever more acute in high-energy physics and cosmology, and is related to the question of non-empirical theory assessment that we have mentioned earlier [43]. See also [70–72] for contemporary views on this problem.

So who was right between Popper and Kuhn? There have been arguments in both directions. The two most influential reactions to the Popper versus Kuhn debate were embodied by Lakatos and Feyerabend, which we will briefly discuss in turn.

Imre Lakatos [74] tried to reconcile Popper's and Kuhn's views. He replaced Kuhn's "paradigm" by the concept of "research programme", which consists of a hard core of fundamental assumptions, and a series of auxiliary hypotheses. These auxiliary hypotheses can serve to increase the predictability of the research programme, or to save it from threats. If most auxiliary hypotheses belong to the first category, and most of the novel predictions are confirmed, then the research programme is in a progressive state. If most predictions of the theory are refuted, and auxiliary hypotheses are invoked to save the research programme, then the latter is degenerative. Research programmes are thus not falsified in Popper's naive sense, but they should be abandoned if they have entered a degenerative state, and a progressive alternative is available which has a stronger empirical content. In this way, Lakatos tried to reframe Kuhn's revolutions on a rational basis, by giving concrete criteria for switching allegiance between research programmes, while updating the essence of Popper's falsification idea.

Feyerabend, on the other hand, dismissed both Popper and Kuhn's views on science (and therefore also Lakatos' attempt at a compromise) [75]. According to Feyerabend's "epistemological anarchism," any standard of rationality or universal methodological rule would be too restrictive and, if really applied, in fact hinder science. Feyerabend concludes, based on a historical analysis, that all commonly accepted rules of science are frequently violated, and that new theories are accepted not because of accord with some universal "scientific method", but because its supporters made use of any "trickery"—apart from rational argumentation, Feyerabend mentions propaganda, psychological tricks, and rhetoric, including jokes and *non sequiturs*—to advance their cause. Illustratively, Feyerabend disagreed with the commonly held negative attitude towards ad hoc hypotheses. In Feyerabend's opinion, ad hoc hypotheses are often required to temporarily make things work until a better understanding is achieved. Furthermore, Feyerabend rejected consistency as a criterion for theory-building, since new theories cannot be expected to be as consistent as the old theory they purport to replace. Feyerabend also made controversial claims about the ideological totalitarianism of science, and its negative impact on (western) society. Few scientists and philosophers of science would probably agree with Feyerabend's most radically relativist claims. Nevertheless, the essence of Feyerabend's analysis has withstood criticisms. As a consequence of Feyerabend's work, together with a more general shift of focus within the philosophy of science, the goal of formulating a universal logical-methodological framework for science, or a single and absolute demarcation criterion between science and non-science, has been mostly abandoned.

However, one further attempt at formalizing the progress of science should be mentioned, namely Bayesian epistemology, or simply Bayesianism. Bayesianism relies on Bayesian inference, and especially its so-called "subjective" interpretation. This is historically prior to the Popper–Kuhn–Lakatos–Feyerabend discussion, but has become widely popular as an approach to scientific progress more recently, and in particular is often mentioned in the context of theoretical physics. Bayes' famous formula which relates conditional probabilities can be written as

$$P(H \,|\, E) = \frac{P(E \,|\, H) \cdot P(H)}{P(E)} \tag{1}$$

where, in this context, $H$ represents a hypothesis and $E$ the evidence, and probabilities are taken as expressing a priori ($P(H)$) and a posteriori ($P(H \,|\, E)$) degrees of belief, and $P(E) = \sum P(E \,|\, H_i)P(H_i)$ the overall probability for the evidence $E$ to actually occur. In practice, only a limited range of hypotheses are summed over, and $P(E)$ becomes a somewhat subjective assessment. The Bayesianists' claim is that Bayes' formula provides a useful model for scientific progress in general [76]. This is certainly true within Kuhnian phases of "normal science". Bayesian inference is commonly and successfully used to assess, for instance, the significance level of the outcomes of particle detector experiments or of cosmological observations. When it comes to paradigm shifts, there is a certain

debate about Bayesianism [77]. The right-hand side in (1) contains the a priori or subjective belief $P(H)$ in the hypothesis $H$. Then, no matter how rigorously one defines $P(E \mid H)$, and thus no matter how rationally the belief in $H$ is updated, the left-hand side will still be a subjective belief, not a proof for the validity or degree of probability for the truth of hypothesis $H$. However, two key points stressed by Bayesianists are the following. First, the subjectivity of the prior beliefs can be avoided by working with Bayesian likelihood factors $B_{12} = \dfrac{P(H_1 \mid E)}{P(H_1)} \left( \dfrac{P(H_2 \mid E)}{P(H_2)} \right)^{-1}$, representing the relative support of evidence $E$ for $H_1$ with respect to $H_2$. These do indeed not depend on the prior beliefs $P(H_i)$, although they still depend on the $P(E \mid H_i)$, for which an agreement among defenders of different paradigms might be equally hard to achieve. Second, when there is a sufficient accumulation of evidence in favour of a particular hypothesis $H$, all rational observers will eventually agree on a high probability for $H$, regardless of their original degree of belief.

### 4.2. Application to the Multiverse

Let us now see how all these general philosophical arguments relate to the multiverse.

### 4.2.1. Popper

From a Popperian point of view, "the multiverse" as a generic theory cannot be falsified, and this probably forms the most frequently heard criticism on the multiverse. However, two nuances are immediately in order.

First, as indicated before, Popper's falsificationist programme is no longer seriously upheld within the philosophy of science, certainly not in its naive form: falsification by itself is not the motor of scientific progress. This indicates that, within theoretical physics, we should overcome the very popular discussions about the multiverse and the anthropic principles centred on falsificationism, with one side defending it [78] and the other side claiming that it is about time to throw it overboard [79]. However, the question of falsifiability is not just about "scientific methodology". Popper's insistence on the testing of theories is still as relevant as it was a hundred years ago. You can formulate theories about reality, but if reality disagrees, the only way for nature to "kick back" and tell us whether our theories are tentatively right or wrong is through empirical confrontation with experiment and observation. Said in other words, a scientific theory should be able to make predictions which are testable: it must be possible to formulate what should be the case empirically (at least in principle) if the theory is true, and in which empirical case the theory should be considered as being in trouble. The idea that this combination of verification and falsification is an essential element in the "scientific method" is still largely uncontroversial among the immense majority of scientists and philosophers of science alike. Even [43], which advocates strongly for "non-empirical theory assessment" based on the alleged success of string theory, admits that such non-empirical assessment should ideally be temporary and that "empirical testing must be the ultimate goal of natural science".

A second nuance is that, although the multiverse in general cannot be falsified, this does not mean that concrete multiverse scenarios cannot be falsified. In fact, in our opinion this is perhaps the most crucial challenge for the multiverse in the near future: to work out concrete scenarios with concrete predictions that could be tested, at least in principle. We will discuss this in more detail in Section 6.

### 4.2.2. Kuhn

Does the multiverse really represent a "paradigm change" or a "scientific revolution", in Kuhn's vocabulary? This question is hard to answer for several reasons. One is that, historically speaking, such paradigm changes are usually not identified as and when they occur, but only a posteriori. Also, despite Kuhn's insistence on the revolutionary character of such paradigm changes, the moment when this revolution has taken place is very hard to pinpoint exactly. The revolutionary process depends on a confluence of several factors. That a single event and/or a single scientist is afterwards highlighted has often more to do with good story-telling than with the real complicated process that has taken place.

Special Relativity is a good example. While often presented as Einstein's first stroke of genius out of the blue, Einstein's treatment was in fact the culmination of a long process with crucial contributions from Lorentz and Poincaré.

These observations are in stark contrast with some messages in the multiverse literature which prophesy a revolution in our understanding of reality (see the examples [4,5,23,24] given earlier). In our opinion, one should be careful with this kind of claims. Announcing a scientific revolution while it is supposedly taking place runs a serious risk of sounding hollow. We are not aware of any research on the frequency of such claims in physics, but it might be interesting to note what is happening in other areas of research. In medical research, for instance, it was found that the frequency of announcements of "unprecedently innovative groundbreaking" ideas has increased up to 15,000% over the past four decades [80]. The authors dryly remark that "whether this perception fits reality should be questioned". The editorial [81] concludes that "it is time to acknowledge that the misrepresentation of research findings through exaggeration or hype is a grave matter for scientific integrity". Despite Kuhn's insistence on the social character of scientific truth-building, for a scientific revolution to take place, it is not sufficient that a particular scientific community claims that it is taking place. Similar feelings have often existed, even in the relatively recent past, and were proven to be wrong much more often than they were right. The development of quantum mechanics is an interesting exception; Chew's bootstrap model, early versions of supergravity and geometrodynamics, and Euclidean quantum gravity are just a few confirming examples.

A related question is: which paradigm is the multiverse supposed to be replacing? The paradigm of "the universe"? Or merely the standard ΛCDM model of cosmology? In the first case, it certainly seems a bit early to argue that "the universe" is in a state of crisis. In the second case: it is true that there are serious puzzles in cosmology, from the nature of dark matter and dark energy to the connection between primordial fluctuations and large-scale structure formation, to name but a few. However, these are challenges for any cosmological model rather than clear-cut problems with the current ΛCDM paradigm. ΛCDM can be interpreted as the simplest cosmological model based on General Relativity which is in agreement with the firmly established bulk of current observations, i.e.: a concordance model with much room for further modifications and extensions. There is at present not a single observation that points towards the assumption of a single universe as the crucial cause of these puzzles. Also, let us not forget that observational cosmology has grown in only a few decades from a phenomenon almost on the margin of science to a blooming area of research, but is still in its infancy. Depending on how one wishes to look at it, one might therefore argue that the ΛCDM model is suffering from serious anomalies, or that it has so far been of an unprecedented success in the history of cosmology. Either way, ΛCDM will undoubtedly require corrections, perhaps even major revisions [82,83]. However, from a purely observational point of view, the case for giving up trying to explain cosmology within a single universe is currently rather thin.

### 4.2.3. Lakatos

According to Lakatos' criterion, a research programme should be abandoned when it enters into a degenerative state, and at the same time a progressive alternative is available. Recall that a research programme consists of a hard core, which is maintained unaltered until the research programme is completely abandoned, and a series of auxiliary hypotheses. A degenerative research programme is characterized by the formulation of auxiliary hypotheses in order to save it from failures to predict or explain empirical observations, while a progressive one uses auxiliary hypotheses to strengthen its empirical content.

From this point of view, most approaches to the multiverse should probably not (yet) really be classified as a research programme. Rather, in Lakatosian terms, the multiverse is in fact an auxiliary hypothesis which has arisen within various existing research programmes (such as string theory and inflation theory) to justify their lack of empirical success, and more in general: the lack of empirical success of any approach to quantum gravity and/or Planckian physics. We do not wish here to jump

to the conclusion that these are all degenerative research programmes. However, it might be relevant to realize that "the multiverse" in its current state does not fit the Lakatosian description of a (mature) research programme.

Moreover, as we already indicated in the Kuhnian discussion above, there is no degenerative research programme in need of replacement (yet). It is certainly true that there are many challenges for the standard ΛCDM model of cosmology, in particular the cosmological constant problem. However, concluding that ΛCDM is in a degenerative state would be a bit overhasty, and at present there is not a progressive alternative with a stronger and more successful empirical record available. With regard to the cosmological constant problem (see also Section 5), the multiverse offers one type of solution, but there also exist several other categories of interesting ideas that do not require positing a multiverse [84], and it is probably fair to say that none of these proposals, neither universe nor multiverse-related, are currently generally accepted as satisfactory.

### 4.2.4. Feyerabend

Within Feyerabend's vision, it is tempting to highlight that some scientists make use not only of rational argumentation, but also of the various types of "trickery" mentioned by Feyerabend to reinforce the impact of their model. There is a certain truth to this: the multiverse cause is omnipresent, especially in the popular scientific press, but also in the academic literature (with a high publication rate of scientific articles). On the positive side, this illustrates that theoretical physicists are no longer isolated in their academic ivory towers, but make an effort to reach out to the general public and present current ideas about fundamental issues to a wider audience. On the negative side, in the absence of empirical testing, despite the robustness of the underlying theories, criticists might argue that the truth-claims of the multiverse rely mainly on a social consensus.

There is a related point for which we will return to Lakatos' terminology: research programmes define which paths to pursue (positive heuristic) but also which paths to avoid (negative heuristic). Is there a real risk that the increasing influence of multiverse ideas might lead to a gradual decline in explorations of alternative approaches in cosmology and high-energy physics as was argued in related contexts [85–87]? The current situation in cosmology does not seem so alarming. In addition, in order to raise a new issue it is necessary to explore its possibilities. We will therefore not further examine this question here. It should be clear that we agree on the importance of empirical testing, and will therefore insist that this should be crucial also within multiverse approaches.

### 4.2.5. Bayesianism

The main weakness, in our opinion, of a Bayesian defense of the multiverse, is the following. Bayesianism is very well-suited to formulate logically how the probability for the true occurrence of a certain event should be updated in the light of observational evidence, but is more questionable when it comes to formalizing paradigm changes.

We pointed out earlier—see Equation (1)—that a sufficient accumulation of evidence in favour of a particular hypothesis $H$ will "force" all rational observers to assign a high degree of probability for $H$. However, it is equally true that anybody who is strongly unconvinced a priori of the hypothesis $H$ will (and, rationally speaking: *should*) refuse to admit a strong probability for the truth of $H$ until there really is overwhelming evidence. In addition, such overwhelming evidence, in the opinion of the large majority of scientists, should still come from the confrontation of the hypothesis with observation. Once such "traditional" scientific proof in the form of empirical verification becomes available, then it is largely irrelevant whether one abides by Bayesian principles or not. It is therefore hard to see how Bayesian inference could formalize scientific progress across the kind of paradigm shift which is currently being defended by some proponents of the multiverse, namely one based largely on theoretical arguments. More generally, despite the unquestionable value of Bayesian inference, the Bayesian view on scientific progress in general (including theoretical paradigm shifts) is a bit

curious, because it represents a return to an attempt at a logical formulation of the progress of science, disregarding the historical evolution from Popper to Feyerabend that we have tried to outline earlier.

We will here briefly sketch three further problems related to Bayesianism.

(1) The first problem is well known as the measure problem. Our universe provides us with a sample of fixed size $n = 1$. This means that almost all statistical properties of the alleged multiverse population are ill-defined, unless one somehow defines a concept of measure across the multiverse population. This can be done essentially in two (interrelated) ways. The first way is to assume some simple distribution, such as a uniform distribution of the possible cosmological constants [88] (or of a small set of variables, for example the cosmological and gravitational constants). However, apart from the fact that it is hard to justify a priori why precisely these variables should characterize the distribution, one should also realize that, with a uniform distribution, many statistical characteristics are essentially determined by the limiting values. Just like in the famous German tank problem, estimating these limiting values based on a single observation entails a very high degree of uncertainty. A second method to define a measure is to assume some fundamental theory, typically string theory [36], and use the theoretical knowledge obtained from this theory to derive a measure. However, this has various associated risks. As pointed out by Ellis [87], "the statistical argument only applies if a multiverse exists; it is simply inapplicable if there is no multiverse: we cannot apply a probability argument if there is no multiverse to apply the concept of probability to." Even if the multiverse really does exist, there is still a risk of circularity: to construct a measure from an empirically unverified theory based on an $n = 1$ sample needs some auxiliary hypothesis, the most obvious possibility being related to what has become known as Vilenkin's mediocrity principle [27], namely that the sample lies in the densest part of the probability distribution. If such a measure can then be constructed, it should obviously show that the $n = 1$ sample indeed lies in the densest part of the probability distribution. However, apart from the almost tautological character of this construction, there is no way of empirically contrasting the obtained measure, not even in principle, since we are by definition limited to the $n = 1$ sample size. This does not deny the adequacy of a Bayesian treatment based on relative likelihoods, which can be useful to compare different multiverse scenarios, for example to determine whether certain parameters or observations favour one multiverse scenario over the other, see e.g., [89]. However, there is no well-defined mechanism of correcting the original theoretical assumptions themselves. So apart from the question of which measure is most adequate, there also exists a challenge of understanding how such a construction based on an $n = 1$ sample could help us in deciding whether a multiverse scenario is really needed, rather than a single universe.

(2) A second problem has to do not so much with Bayesianism in itself, but with naive applications of it. Polchinski famously arrived at a 94% probability for the multiverse to exist [90,91]. Polchinski's estimate is based on four yes–no questions (e.g., is there a satisfactory understanding for the cosmological constant value?). Since conventional (non-multiverse) physics answers four times "no", Polchinski arrives at a probability of $1 - (1/2)^4 = 0.9375$ in favour of the multiverse.[4] Let us play the devil's advocate and, for the mere sake of the argument, defend the creationist view of intelligent design in biology. State any four gaps in the evolutionary picture of life and humanity. The Polchinski–Bayesian conclusion would be that there is a 93.75% probability that the universe was literally created by God in seven days. If this argument sounds too far-fetched, let us insist on the key point: the current lack of explanation for any scientific challenge within conventional single-universe relativistic cosmology in itself is not a sufficient support for an anthropic or multiverse argument. We will come back to this question in Section 4.3.

(3) The third problem is closely related to the previous one, namely the risk of accepting Bayesianism in combination with purely theoretical arguments as a substitute for empirical testing.

---

[4]　In reality Polchinski's four questions are not independent, so the numerical estimate is incorrect even from a purely probabilistic point of view. However, since Polchinski himself states that the number itself is not important, we will not further dissect this issue.

This is best illustrated by an example. According to a Bayesian reasoning with purely theoretical arguments, there should have been almost 100% certainty in favour of the Georgi–Glashow SU(5) model [92]: this was closely based on some of the best physical theories that mankind has ever produced, it was mathematically elegant and favoured by a large proportion of theoretical physicists, and no alternatives even closely as appealing were available at the time. Yet, proton decay was not observed and so the Georgi–Glashow SU(5) unification model turned out to be wrong. This again illustrates our continuous insistence on empirical assessment.

*4.3. Consistency and Uniqueness Claims*

In the previous section, we have insisted on empirical theory assessment. Within the multiverse context, some authors propose to diminish the importance thereof, and to replace it (partially) by purely theoretical criteria. This is another line of thought where the philosophy and history of science can be relevant.

The idea that theoretical arguments, for example criteria of mathematical consistency and elegance, can illuminate the path towards a correct "fundamental" theory, is not new. On the contrary, this is closely related to Platonism, one of the oldest branches of western philosophy. It has resurged in theoretical physics repeatedly, especially in the past century or so [93]. The common pattern is striking: a scientist or group of scientists believes in the fundamentality and finality of the theory they are working on, based on the past success of the building blocks of this theory and the elegance of their construction. Empirical predictivity is either looked down upon, or the lack of empirical success is simply disregarded. Eventually, so far at least, the theory turns out to be either completely wrong or, in the best of cases, simply void of empirical content. The best-known example is perhaps Descartes' vortex theory, abandoned in favour of Newton's empirically much more successful laws of motion. However, more recent examples also abound. A ring-vortex theory, highly popular in late 19th century Britain, was developed in quite some mathematical detail by such famous contributors as William "Lord Kelvin" Thomson and FitzGerald. Although it never managed any level of empirical success, it continued to be defended by many scientists for several decades because of its elegance. As another example, Eddington developed a fundamental "Relativity Theory of Protons and Electrons" based on the construction of a series of fundamental constants which were supposed to relate microphysics and cosmology. In Eddington's view, the truth of his theory followed from purely epistemological considerations. Empirical confirmation was completely secondary, and even though he did in fact make quite a few observational predictions, he would simply disregard any disagreement with actual observations rather than let them ruin his beautiful theory. Many more historic and contemporary examples are discussed in detail in the excellent [93].

While [93] cautiously avoids extracting any explicit conclusions with regard to the current situation, authors such as [94,95] have argued that a blind quest for mathematical beauty has indeed led contemporary fundamental physics astray. So does history simply repeat itself? Let us examine some reasons to believe that the case of the multiverse might be different. From a social point of view, the current state with respect to various approaches in quantum gravity represents the first time that a large and international scientific community defend a common idea based on such theoretical criteria. The previous occurrences were mainly of single scientists (including such prestigious ones as Eddington and his "Fundamental Theory" or even Einstein and his "Unified Field Theory"), or had at most "national" success (the late 19th-century vortex theory, which was called a "Victorian theory of everything" by Kragh [96], was very popular in the UK but had limited resonance in the rest of the world). With respect to scientific content, the key argument in string theory and some multiverse-related approaches is that the theoretical "gap" to be bridged is shallow, in other words: that the multiverse is a natural continuation of our best theories, general relativity and quantum field theory; that we are indeed close to finding such a "final theory", and that consistency, elegance and uniqueness should therefore be sufficient arguments to solve the remaining problems (until the solution is eventually confirmed empirically). In this context, there is a statement by Popper that comes

to mind: "Whenever a theory appears to you as the only possible one, take this as a sign that you have neither understood the theory nor the problem which it was intended to solve." [97].

Let us try to make a more precise counter-argument.

First of all, it is true of course that unification has been an important motor in the history of science. However, unification and uniqueness are two different concepts. Their apparent relation finds its origin in the reductionist idea that gradual unification will lead to a unique theory of everything, at the top of a pyramid of theories. This idea was strongly criticized by some physicists [98] and philosophers [99] alike, see also [100], who argue that the major advances in fundamental physics in the recent past have relied on a combination of unification and emergence.

Second, it is an interesting question whether the current state of physics, and in particular general relativity and quantum field theory (the precursors of the multiverse), could have been achieved through arguments of consistency, elegance and uniqueness. For general relativity, such arguments have certainly been crucial in Einstein's reasoning, and so one might be tempted to answer "yes". However, for quantum field theory, and its application to particle physics, although we cannot repeat history to answer the hypothetical question, the historical answer is a definite "no", as described in detail for the case of quarks in [73].

Third, the final unification is believed to take place at the Planck scale, and so the "dreams of a final theory" [101] are related to the idea that we are close to uncovering Planckian physics. This third point deserves a more detailed analysis, which we will undertake in the next section.

## 5. Fine-Tuning and the Multiverse... or Is It Really a Tale of Scales?

One of the strongest arguments in favour of the multiverse is the cosmological constant problem. Since the observed value of $\Lambda$ is some 120 orders of magnitude smaller than the straightforward theoretical estimation[5], and any value of $\Lambda$ very different from the actually observed one would probably make life in the universe impossible, it is argued that "the only known way to address [this problem] without invoking incredible fine-tuning [is] related to the anthropic principle, and, therefore, to the theory of the multiverse" [5].

Let us jump back to the Planck-scale unification argument of Section 4.3 for a moment. Some scientists working in quantum gravity believe that we are close to uncovering Planck-scale physics, and that consistency and perhaps uniqueness arguments should therefore be sufficient to bridge the remaining gap towards a final theory [43,90,101], possibly a multiverse theory.

The highest-energy physics that we actively control is the energy produced at the LHC. This is currently on the order of 10 TeV, i.e., $10^4$ GeV. Compare this to the Planck scale, $10^{19}$ GeV, the scale at which quantum gravity supposedly take place. There is a difference of 15 orders of magnitude. Even high-energy cosmic ray detection rarely exceeds $10^4$ TeV, still 13 orders of magnitude below the Planck scale. This problem is of course well-known among high-energy physicists, but there seems nevertheless to exist an optimistic view on bridging this gap [90]. However, two simple comparisons might serve as a cold shower. The extrapolation from the highest-energy physics that we control empirically to the physical theories which justify the idea of a multiverse is (literally) still several orders of magnitude stronger than the extrapolation from a grain of salt (size $10^{-4}$ m) to the size of the moon (diameter $10^6$ m). To put another example: imagine that a biologist would claim that,

---

[5]    Just in case some reader might benefit from a reminder, the theoretical estimate comes essentially from assuming that the cosmological constant represents the vacuum energy $E_{vac}$, imposing a cut-off $k_c$ to the theory and calculating $E_{vac}$ by integrating over all degrees of freedom up to $k_c$, which gives $E_{vac} = \hbar k_c^4$ (a result which is consistent with a straightforward dimensional analysis [102]). Assuming $k_c = E_{Planck}$ immediately leads to the undesired result, while even $k_c = E_{EW}$, with $E_{EW}$ the electroweak scale, still leads to a discrepancy of some 50 orders of magnitude. It might be worth insisting that it is essential to insert a cut-off in the calculation in order to avoid an even more unpleasant prediction for the vacuum energy, namely infinity. Note that the observational energy scale associated with dark energy is in fact small, and might therefore be due to quantum field effects potentially accessible to near-future observations. However, this would still leave the cosmological coincidence problem unexplained, namely why the matter energy density and the dark energy density have the same order of magnitude in the present epoch.

by studying the macroscopic properties of the largest living beings on earth, blue whales, he could infer the biological structure of the smallest known bacterial cells, with sizes 0.1 μm. The mere scale difference of $10^8$ is peanuts in comparison with the jump from the LHC to the Planck scale.

There are strong theoretical reasons to believe that the cosmological constant is related to Planck-scale physics. In fact, the argument in favour of the landscape of string theory rests precisely on Planck-scale arguments. Therefore, because of the energy gap just described, perhaps we should simply admit that the "worst theoretical prediction in the history of physics" is due to our (unsurprising) ignorance of physics at the Planck scale, that we are currently exploring many ideas, but that all of these (including the multiverse) are so far still in an embryonic state.

It is tempting to put the blame on the lack of empirical data [43]. It is of course true that there is no empirical data available for physics at the Planck scale. However, two interpretations are possible. One could say that experimentalists have not been able to keep up with theorists. Perhaps a fairer interpretation is that, after the enormous success of quantum field theory and the standard model of particle physics, theorists have run ahead, jumping several scales and constructing theories well above current experimental possibilities. As we have defended above, scientific progress typically rests on a complex interplay between theory and observation, and this might be even more important as we move further and further away from direct sensory experience. Bottom-up and top-down approaches in fundamental physics should be complementary [103]. Presently the equilibrium in the search for quantum gravity is a bit distorted.[6] Near-future observational surveys with respect to the "dark sector" of the universe such as DESI and EUCLID are promising. However, because of the scale problem that we have just stressed, the key message should probably be one of patience and of anticipating slow and indirect progress, rather than immediate spectacular advances.

## 6. Physical Multiverse and Testability

The previous discussion leads to the following general issue: How could the overall conceptual challenge be met of converting the multiverse from a speculative (or even metaphysical) consideration into a physical theory? The multiverse currently provides an interesting framework to understand reality, but it should also be followed by testable predictions. To come back to the relation between the multiverse idea as an epistemological extension of the Copernican revolution that we mentioned at the beginning of Section 3: Humanity has gradually realized its loss of importance as the center of existence when science has been able to look further and realize that those observed objects were in fact other structures similar to ours: other planets, other stars, other galaxies. The possible extension to the multiverse is not accompanied by any such direct observation, and is therefore of a different speculative order. Ultimately, the multiverse question comes down to determining whether, in order to confront the observational challenges and anomalies of cosmology, it is sufficient to consider a single universe, or whether we need a multiverse scenario. The main element in the effort to answer this question consists of setting up multiverse scenarios and looking for empirical predictions which can be tested. Depending on the type of multiverse scenario, this can be very hard, perhaps even impossible. For instance, in a multiverse scenario where the different universes possess different physical laws or mathematical structures, it is hard to see how to look for interactions with our universe. Alternative ways of assessment might then still lead to a certain degree of confidence, but always provided that other parts of the theory can be tested empirically.

In this section, we want to sketch a possible way of approaching the empirical multiverse question, i.e.: to establish empirical predictions in the traditional physical sense, limited to our universe but nevertheless allowing us to find some hint of interactions with other universes. In order to describe

---

[6]   The only well-developed bottom-up approach to "quantum gravity phenomenology" is the ongoing search for Lorentz Invariance Violations [104,105]. However, it might be useful to stress that neither string theory nor loop quantum gravity make clear and unambiguous predictions about Lorentz Invariance, not even at a qualitative level.

such "physical multiverse" scenarios, no specific multiverse model is required. It is sufficient to impose certain minimal requirements on the multiverse scenario.

The first of these requirements is the classical independence of the spacetimes of each composing universe, at the level of the standard phenomena in General Relativity. This degree of independence is essential in order to consider each universe as a separate and differentiable entity. In the opposite case, it would be possible to define the different components as different regions of a single universe. If we understand as standard spacetime connections the ones allowed by General Relativity (without introducing exotic issues such as closed timelike curves) we assume that such causally completely determined relations do not exist between the spacetimes of the different universes that compose the multiverse. Note that the existence of non-standard (classical or quantum) connections is not prohibited by this definition. On the contrary, these are essential in order to have some possibility of interaction among universes and therefore some empirical imprint to look for.

The second requirement is that each of the universes must be potentially observable, by direct or indirect measures, from some other universe. In this way, physical predictions in our universe can be established as a consequence of the physics of the whole multiverse. This shows that the independence among universes works only at the (classical) level of the spacetimes per se, not of all its components. There must exist some degree of interaction among them.

Finally, we also impose that, if the constants of nature are allowed to vary from one universe to another, then the values of these constants must be linked through the physical laws governing the overall multiverse. In other words, these physical constants cannot emerge independently but must be correlated, for instance, through quantum entanglement effects between universes.

One could paraphrase these three conditions by saying that the different universes in the multiverse should have causally independent spacetimes but with correlations among them. These requirements do not impose any specific physical scenario, but are sufficient to define testable consequences of any multiverse scenario which obeys them.

In order to clarify this concept one can consider a classification of these correlations in terms of their classical or quantum nature. The classical correlations could be given, for instance, by considering the multiverse as a multiply connected spacetime, where each universe is connected with other by means of Lorentzian tunnels [106,107]. It is important to note that, from the first condition given above, there cannot exist causal relations among the different universes. The existence of causal relations would entail the existence of a common time between both universes which could then not be considered independent. The connections could therefore be formed by wormholes converted into time machines, providing closed timelike curves in the interior of the tunnel [108]. Potential observable effects of wormholes were studied in several papers [109–114]. The current challenge in the multiverse context is the search of an unequivocal observable effect of such a wormhole connection with another universe [115].

Quantum correlations could come from the quantum entanglement between universes. According to the first physical multiverse requirement, the spacetimes of each component universe must be classically differentiable. However, in this scenario they would not be quantum separable, giving rise to an entangled multiverse [52]. In this context, one could determine the effects on our universe that show up as a consequence of these inter-universe quantum correlations. In the absence of a classical channel, these correlations cannot be directly detected. However, the influence of such correlations can be examined, for example on the value of the cosmological constant. It might be hard to imagine such an indirect effect which could not equally be explained within a single-universe scenario. However, the study of the different schemes of interaction among universes and the development of a toy-model catalogue of observable effects and predictions allows an important progress towards multiverse phenomenology and of the types of effects that could be expected in more detailed scenarios [52,53,56–58]. The investigation of these quantum correlations is complicated by the lack of a quantum theory describing our universe, and most current models therefore focus on qualitative approximations to the collective phenomena that can arise in the consideration of a

quantum multiverse. Alternatively, an exhaustive description of the spacetimes in the framework of quantum mechanics could be attempted. This allows a more rigorous description of the interactions, but at the cost of a great technical complexity which limits the conception of different universes [116].

We are still very far from having a complete theory that would allow us to settle the present discussion, or a fully systematic way of deriving empirical consequences from concrete multiverse scenarios. Nonetheless, the physical multiverse ideas just described show that, at least for certain classes of multiverse models, it should be possible to extract empirical predictions based on relatively general considerations. So it is interesting to keep them in mind when dealing with these issues. Also, if such inter-universe correlations as just described really exist, then this would indicate the necessity of considering the multiverse as an indivisible framework. This in itself should be sufficient motivation to construct such generic physical multiverse models and study their possible empirical effects.

## 7. Conclusions

In our view, it is not so important to determine whether speculations about "the multiverse" are part of science or not. Only time will tell. However, multiverse scenarios should certainly be recognized for what they (still) are: an embryonic framework which can be useful to understand and formulate certain problems related to cosmology, but which is still far away from being testable in any general physical sense. Care should perhaps be taken with claims that "multiverse theories are utterly conventionally scientific" [117], or that Bayesian arguments show that, by a "conservative" estimate, "the likelihood that the multiverse exists [is] 94%" and that "those who find this calculation amusing (...) should be a bit more humble" [90]. Such claims might be more counter-productive than anything else. Indeed, they do not fairly reflect the current scientific status of the field, nor do they agree with historical and philosophical analyses and in particular the importance of empirical content and of the complex interplay between theory-building and observation required for scientific progress.

Vice versa, despite all the warnings that we have formulated, it is certainly not our intention to dismiss the general idea of a multiverse as a developing physical theory. This would imply closing the door to a whole range of ideas and techniques that are currently being developed, and some of which could indeed turn out to be fundamental in our understanding of the nature of spacetime. However, empirical testing should always remain the central aim of science, even (or perhaps: especially) in the multiverse epoch. In that sense, the general framework for a physical multiverse that we have discussed could be useful as a guide for the development of empirical multiverse scenarios, and more generally: to discriminate emergent ideas and to look for the possible testability of different cosmological scenarios involving either a single universe or a multiverse in any of the various multiverse definitions.

**Author Contributions:** Both authors contributed equally to this work.

**Funding:** The authors acknowledge Project No. FIS2017-86497-C2-2-P (A.A.-S.) and FIS2017-86497-C2-1 (G.J.) from the Spanish Mineco.

**Acknowledgments:** A.A.-S. wants to acknowledge the memory of Pedro Félix González-Díaz as a source of inspiration for her research and for originating the discussion about the physical multiverse.

**Conflicts of Interest:** The authors declare no conflict of interest.

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
