# Peer review of "Conceptual Challenges on the Road to the Multiverse"

_universe, doi:10.3390/universe5100212_

Round 1

Reviewer 1 Report

The article discusses unsolved issues with multiverse proposals and the prospects for future development and assessment of such proposals. The work is overall well-argued and well-presented but would benefit from clarifications, and additional discussion to connect to other work in the area and more general philosophical topics that are central for the problem formulation. Below follows comments on each section. 

Abstract

The abstract refers to a "change of paradigm", but what is the existing paradigm and what is the possibly new paradigm is implicit. Making this explicit might be helpful for readers. 

1. Introduction

A good and concise historical overview is given. The authors might consider mentioning the Harlow-Shapley Great Debate and the discovery of other galaxies associated with the modern cosmological revolution, here or at another suitable place in the text, both for historical completeness and as a possibly illustrative recent example of paradigm change. 

2. The multiverse: Nothing new under the suns?

The sentence "In the 18th century, Kant was the first to interpret the astronomical observations of distant nebulous structures as other galaxies, and proposed the existence of habitable worlds around stars other than our Sun" is inconsistent with established historical accounts. While Kant did popularize and argue for the idea, Emanuel Swedenborg appears to be the first to propose other solar systems, and Thomas Wright the first to appeal to observational evidence in support of this idea. 

3. Definition and classification of the multiverse

It is highlighted that clarifications of the physical and mathematical relations between different multiverse scenarios is desirable. It would be vaulable to provide the reader with some examples or ideas for how this might be done more clearly what the status and prospects of this might be, etc. This strikes this reviewer as a key point to highlight further in this work, even though you do address the issue in section 6.

The notion of theory confirmation is discussed. Elsewhere the notions of verification and viability are discussed. The authors ought to reflect on whether a distinction between all these concepts is needed in their argument or not. It would also be advisable to define these concepts explicitly, since readers with different backgrounds may interpret them in ways not intended (the philosophical concept of theory confirmation is notoriously misunderstood by physicists, producing much unnecessary confusion). 

4. Philosophical aspects

It is stated that "But from a purely observational point of view, the case for giving up trying to explain cosmology within a single universe is currently rather thin.".  The authors could offer some indication of why they hold this opinion, and why also it will undoubtedly require corrections. This would also give an opportunity to address the importance of General Relativity and its possible extensions / quantum gravity for the Lambda-CDM model, which otherwise appears as rather monolithic in the discussions of its status in relation to different philosophers' models for scientific paradigm change. 

The general problem of under-determination of scientific theories is not discussed, but would be relevant to bring into the discussion. Both the structure of under-determination (e.g. Dardashti's theory-space structure problem) and approaches to assess under-determination (e.g. Dawid's non-empirical theory assessment, Sahlen's axiological Bayesianism) would be worth mentioning. This also specifically applies to the discussion on Bayesianism. A brief discussion of principles of epistemic justification including "Inference to the Best Explanation" / abduction would also be useful for completeness for a varied readership. 

The discussion of consistency and uniqueness claims would benefit from relating to the debate that has been spurred by Hossenfelder's recent book "Lost in Math". It is also stated in this section that "The common pattern is striking: a scientist or group of scientists believes in the fundamentality and finality of the theory they are working on, based on the past success of the building blocks of this theory and the elegance of their construction. Empirical predictivity is either looked down upon, or the lack of empirical success is simply disregarded. Eventually, so far at least, the theory turns out to be either completely wrong or, in the best of cases, simply void of empirical content." This is stated without examples, evidence or other arguments to back the claim. The argument ought to be strengthened and clarified, as it is the basis upon which the following multiverse assessment discussion rests. 

5. - 7. 

No specific comments; but in view of the above comments, it may be relevant to revise these to bring them in line with an expanded and clarified discussion.

Author Response

Comments and Suggestions for Authors

The article discusses unsolved issues with multiverse proposals and the prospects for future development and assessment of such proposals. The work is overall well-argued and well-presented but would benefit from clarifications, and additional discussion to connect to other work in the area and more general philosophical topics that are central for the problem formulation. Below follows comments on each section.

We kindly thank the referee for his overall assessment and for his many constructive remarks, which we address in turn below.

Abstract

The abstract refers to a "change of paradigm", but what is the existing paradigm and what is the possibly new paradigm is implicit. Making this explicit might be helpful for readers.

In order to make this point clearer, we have changed the sentence to “possible change of paradigm from a single universe to a multiverse scenario”

1. Introduction

A good and concise historical overview is given. The authors might consider mentioning the Harlow-Shapley Great Debate and the discovery of other galaxies associated with the modern cosmological revolution, here or at another suitable place in the text, both for historical completeness and as a possibly illustrative recent example of paradigm change.

We acknowledge the comment of the referee and we have included a mention of this debate in section 2, see lines 56-62.

2. The multiverse: Nothing new under the suns?

The sentence "In the 18th century, Kant was the first to interpret the astronomical observations of distant nebulous structures as other galaxies, and proposed the existence of habitable worlds around stars other than our Sun" is inconsistent with established historical accounts. While Kant did popularize and argue for the idea, Emanuel Swedenborg appears to be the first to propose other solar systems, and Thomas Wright the first to appeal to observational evidence in support of this idea.

We thank the referee for pointing this out to us and have accordingly modified lines 50-55.

3. Definition and classification of the multiverse

It is highlighted that clarifications of the physical and mathematical relations between different multiverse scenarios is desirable. It would be vaulable to provide the reader with some examples or ideas for how this might be done more clearly what the status and prospects of this might be, etc. This strikes this reviewer as a key point to highlight further in this work, even though you do address the issue in section 6.

We absolutely agree with the referee that this is a key challenge, although this seems to be barely acknowledged within the multiverse research community. In fact, although many popularizing articles and books seem to imply that the relation is obvious (at least conceptually, though perhaps not technically), we are not aware of any scientific article tackling the question seriously from a technical side. We have added a discussion of this point in lines. 174-190.

The notion of theory confirmation is discussed. Elsewhere the notions of verification and viability are discussed. The authors ought to reflect on whether a distinction between all these concepts is needed in their argument or not. It would also be advisable to define these concepts explicitly, since readers with different backgrounds may interpret them in ways not intended (the philosophical concept of theory confirmation is notoriously misunderstood by physicists, producing much unnecessary confusion).

We agree with the referee that these are delicate issues which have a serious impact on high-energy physics. However, we hope that the historico-philosophical discussion which we have sketched in section 4 should be sufficient to convince the interested reader that these are delicate issues indeed, while not distracting the less philosophy-inclined physicist from our main argument, and so we prefer to leave them “undefined” in section 3. Nevertheless, we have added a reference to section 4 in line 201 when we first mention “non-empirical theory confirmation”.

4. Philosophical aspects

It is stated that "But from a purely observational point of view, the case for giving up trying to explain cosmology within a single universe is currently rather thin.". The authors could offer some indication of why they hold this opinion, and why also it will undoubtedly require corrections. This would also give an opportunity to address the importance of General Relativity and its possible extensions / quantum gravity for the Lambda-CDM model, which otherwise appears as rather monolithic in the discussions of its status in relation to different philosophers' models for scientific paradigm change.

We have expanded the corresponding paragraph to make the essential point clear, namely that although there are indeed many puzzles in present-day cosmology, there is not a single observation that pinpoints the assumption of a single universe as a cause of these puzzles. See lines 404-413.

It is true that extensions (and possibly modifications) to GR are likely to play an important role in future cosmology, and in modifications to Lambda-CDM. And we agree that philosophical perceptions of GR (and more generally of spacetime: essential? emergent?…) might play an important role in this debate. For example because the models for primordial fluctuation generation and/or cosmological bounces are closely related to quantum gravity models, which have very different views on the status of GR. However, we feel that this would take us beyond the scope of this paper.

The general problem of under-determination of scientific theories is not discussed, but would be relevant to bring into the discussion. Both the structure of under-determination (e.g. Dardashti's theory-space structure problem) and approaches to assess under-determination (e.g. Dawid's non-empirical theory assessment, Sahlen's axiological Bayesianism) would be worth mentioning. This also specifically applies to the discussion on Bayesianism. A brief discussion of principles of epistemic justification including "Inference to the Best Explanation" / abduction would also be useful for completeness for a varied readership.

Once more, we fully agree with the referee: underdetermination of theory by evidence is one of the key issues here, and in fact we had mentioned it in earlier drafts of our manuscript. However, we felt that it was hard to do justice to the question without dedicating a serious amount of space to it, which we preferred to avoid: we might be wrong, but it is our impression that a 15-page article with a strongly philosophical-conceptual tone is about the maximum that most physicists are willing to digest. However, at the referee’s suggestion we have reintroduced a brief mention to underdetermination with some references, see lines 285-289, and hope to discuss this in some depth in further work.

We have preferred to leave “inference to the best explanation” out of our manuscript to avoid the risk of concept-dropping and saturation.

The discussion of consistency and uniqueness claims would benefit from relating to the debate that has been spurred by Hossenfelder's recent book "Lost in Math". It is also stated in this section that "The common pattern is striking: a scientist or group of scientists believes in the fundamentality and finality of the theory they are working on, based on the past success of the building blocks of this theory and the elegance of their construction. Empirical predictivity is either looked down upon, or the lack of empirical success is simply disregarded. Eventually, so far at least, the theory turns out to be either completely wrong or, in the best of cases, simply void of empirical content." This is stated without examples, evidence or other arguments to back the claim. The argument ought to be strengthened and clarified, as it is the basis upon which the following multiverse assessment discussion rests.

We have expanded the discussion with some examples, and connected this with Hossenfelder’s book, see lines 558-561.

5. - 7.

No specific comments; but in view of the above comments, it may be relevant to revise these to bring them in line with an expanded and clarified discussion.

As mentioned in the previous answers, we agree with the referee that several additional (mainly philosophical) issues are relevant in the multiverse context. However, we prefer to reserve these for further work in order not to overload the current manuscript. We therefore believe that no further modifications are required in the remaining sections (although we have made several additional clarifications at the other referee’s suggestion).

Reviewer 2 Report

Referee’s report on “conceptual challenges on the road to the multiverse”

I believe this is a useful paper that clearly layout the conceptual challenges associated with studying the Universe.   I have a number of suggestions for improvement to the paper, but I think the authors will be easily able to adapt the paper.

Introduction.

The paper needs to start off will a clear but brief statement of what is meant by the multiverse, and with a clear distinction why the term “universe” is not sufficient. This can then be tightened up in section 3 and 6, but its not possible to decide if the concepts cited in section 2 really are “multiverse”-like, in the sense that emerges in section 3 etc, rather than just describing a large, but causally connected universe. For example, Wright (see below) and Kant refer to nebulae as “island universes”, they are clearly causally connected to us since they can be seen.  Presumably this rules out this idea as a “multiverse” in the modern sense.

Section 2.

Kant was not the first person to suggest that the spiral nebulae were island universes. He read of the idea from Thomas Wright’s “An Original Theory, or New Hypothesis of the Universe”. This is relevant since Wright was an observational astronomer (and landscape gardener designer!). This helps with the argument of the paper, and the paper should stress that this idea was driven by observational evidence as well as philosophical considerations (he imagined each island universe to have its own creator!). Despite the title, it wasn’t accepted as a “revolution that completely alters our understanding” until Hubble was able to measure the distances to the nebulae using variable stars and hence cause a very real paradigm shift. I think the analogy is a very good one. At the very least, this factual error should be corrected.

Section 3.

This section gets muddied because it presents the cosmological argument for the multiverse in suitable detail, but does not do the same for Everett’s Many World’s interpretation of quantum mechanics. These are really very different concepts that are largely independent. They overlap if the selection of physical parameters at the exit of the Planck regime is a quantum process. However, even if physical constants are invariant, we would still need the Many-Worlds type multiverse to make sense of quantum mechanics.  Expanding the paper to appropriately treat the Many Worlds interpretation would loose the tight focus that the paper currently has. I suggest, therefore, that the authors clearly state that are focusing on the cosmological multiverse and avoid confusion.

The section beginning on Line 95 starts by suggesting “the multiverse encompasses all the multiple possible universes predicted by an underlying theory”.  My concern is that this says nothing about those possibilities being realised: the multiverse is proposed simply as a mathematical theorem. I don’t think this is satisfactory starting point, even as a vague  one. Surely an essential part of the multiverse concept is that these other universes exist in reality.

Line 152. As noted above, The Many Worlds-type of multiverse would still exist. Also, if the inflationary patch in which we live was far larger than our horizon, would we be happy for the causally disconnected regions (which have the same physical parameters) to be referred to as a “multiverse”. I suspect not!  The authors should clarify these points.

Line 182. Can this entanglement really work? In the situations I’m used to, it is only when the two systems are causally compared that the entanglement becomes apparent. So would the entanglement ever become evident to the observers since they can never share their results. The authors should clarify how an observer might become aware of the entanglement.

Section 4.1

The P(E) term appearing in the equation needs to be defined. This is important, since P(E) is normally taken to the be sum of P(E|H) over all possible H. This is what renders the probability of the truth of H subjective. If the two scientists do not agree on what encompasses “all possible” H, they cannot compute the probability of the truth. This mathematical point underlies much of the following text, and Section 4.2.5. Making this clear would make the text crisper and shorter

Section 4.2.2

Line 366. The authors need to expand on why there are “serious frictions between cosmological observations and the LCDM model”. I cannot see this to be true. This contentious statement certainly needs to be backed up by recent citations. It is very true that the nature of the dark matter is elusive, and that the model contains puzzling coincidences, such as the similar abundance of  matter and dark matter, but these are puzzles rather than an issue with the cosmological model. This is much the same way that the parameters of the Standard Model of particle physics are not explained, but the model itself performs well. This all strengthens the point made, however: observational data do not give any reason to call in the multiverse. The multiverse is only driven by a wish to reach a better understanding of the parameter values.

Line 383. This is one of the sections that does not do justice to the Many Worlds interpretation of QM, and its consequent version of the multiverse. It will be better to focus on the cosmological version of the multiverse.

Section 4.2.5

Line 423  This section can be shortened, as noted above.

Line 462. Polchinski’s argument is so naive that it cannot possibly be referred to as Bayesian. Since it is erroneous, I would question why it is worth including.

Line 466. This section is overly negative. While we are unlikely to be able to fully determine the probability of truth, we can focus on the likelihood part of Bayes’ theorem and thus determine whether the parameters of our Universe are likely in a particular Multiverse scenario.  We can then compare scenarios on a relative basis.  This would interpret the “mediocrity principle” as a familiar likelihood ratio test. While the power of such a test is limited in an n=1 sample, the Bayesian method can still be applied and the implications and caveats quantified.  The authors should not dismiss this approach.

This section would benefit from some discussion of the difficulty of estimating P(E|H) based on anthropic arguments. 

A major uncertainty is to decide how similar the Universe needs to be to our own in order that it contains observers that try to measure the cosmological parameters. Dark energy is particularly tricky: is it enough to simply form galaxies? (The weak anthropic principle). Because of recent advances in understanding galaxy formation, this can be calculated reasonably accurately - see Barnes et al., 2018 (MNRAS 477 3727) for an example. A stronger anthropic principle would demand intelligent observers with telescopes capable of mapping the Hubble flow. Some universes will contain only one galaxy per causal patch, so would large telescopes ever be funded?  The authors should include some discussion of the fundamental difficulties with such strong anthropic arguments.

Section 5

The paper quotes Lambda as being 10^120 orders of magnitude smaller than the relevant energy scale. The argument is true, but the origin of the energy could be down to quantum field effects and the matter - dark matter mass asymmetry.

This would put the physics of dark matter (and thus maybe dark energy) with in reach of current accelerators. The energy scale of the dark energy itself is tiny and potentially accessible to desk top experiments. In a few year, the physics of dark matter/ dark energy may well be solved, but this will not easily explain the coincidence of their values. The authors should include discussion of this point. 

Line 568. To finish on a more optimistic note, the forthcoming DESI and EUCLID surveys will measure the growth of large scale structure in the Universe with incredible precision. It is very likely that that these experiments may provide key evidence that drives theorists to a better understanding of (quantum) gravity.

This sections asserts that “there is no empirical evidence”. But there is some, albeit of the n=1 variety. The authors should give this more credence. The issue, however, is that Weinberg’s paper sets quite broad limits on the range of Lambda values that could be compatible with our Universe (in the P(E|H) or mediocrity sense). Buosso et al (2007, PhRvD, 76 3513)  and more recently Barnes et al 2018 look at this issue in more detail. In particular, in Barnes et al  use the latest cosmological simulations to estimate the abundance of galaxies and hence apply a weak anthropic principle to compute P(E|H) for a wide range of Lambda values. They find that our Universe’ value is not at all typical, particularly if the energy argument is used to motivate a log-uniform prior. Lambda values 100x our own still generate abundant stars and galaxies. Disappointingly, the evidence seems to disfavour the Universe (albeit with the n=1 sample caveats). The authors should include discussion of this point.

Section 6

Line 604.  The paper should acknowledge that n=1 statistics might give some support and discuss what further support would be need to be for the evidence to be conclusive.

Line 609.  I am sceptical that the entanglement effects would be apparent to an observer that sees only one universe. The paper should provide a clear example of how this would work without the universes being causally connected in the conventional sense.

Author Response

Comments and Suggestions for Authors

Referee’s report on “conceptual challenges on the road to the multiverse”

I believe this is a useful paper that clearly layout the conceptual challenges associated with studying the Universe. I have a number of suggestions for improvement to the paper, but I think the authors will be easily able to adapt the paper.

We kindly thank the referee for his overall assessment and for his many constructive remarks, which we address in turn below.

Introduction.

The paper needs to start off will a clear but brief statement of what is meant by the multiverse, and with a clear distinction why the term “universe” is not sufficient. This can then be tightened up in section 3 and 6, but its not possible to decide if the concepts cited in section 2 really are “multiverse”-like, in the sense that emerges in section 3 etc, rather than just describing a large, but causally connected universe. For example, Wright (see below) and Kant refer to nebulae as “island universes”, they are clearly causally connected to us since they can be seen. Presumably this rules out this idea as a “multiverse” in the modern sense.

A key point of our first 3 sections is precisely that there is no commonly agreed-upon definition of the multiverse, neither historically nor in contemporary physics. How strictly the concepts cited in section 2 should be judged as “multiverse”-like from the point of view of the present understanding of this concept does not strike us as particularly important. Rather, we discuss them here to sketch the historical evolution of these ideas, and to show that -contrarily to what is often claimed, especially in popularizing accounts- the multiverse idea is not especially revolutionary, but has evolved historically in a more or less smooth way.

We have added a sentence at the end of the first paragraph of the section, see lines 17-21.

Section 2.

Kant was not the first person to suggest that the spiral nebulae were island universes. He read of the idea from Thomas Wright’s “An Original Theory, or New Hypothesis of the Universe”. This is relevant since Wright was an observational astronomer (and landscape gardener designer!). This helps with the argument of the paper, and the paper should stress that this idea was driven by observational evidence as well as philosophical considerations (he imagined each island universe to have its own creator!). Despite the title, it wasn’t accepted as a “revolution that completely alters our understanding” until Hubble was able to measure the distances to the nebulae using variable stars and hence cause a very real paradigm shift. I think the analogy is a very good one. At the very least, this factual error should be corrected.

We thank the referee for pointing this out to us, and agree that the interplay between philosophical idea and observation is a nice example of a succesful paradigm change. We have changed and elaborated the sentence, see lines 50-55.

Section 3.

This section gets muddied because it presents the cosmological argument for the multiverse in suitable detail, but does not do the same for Everett’s Many World’s interpretation of quantum mechanics. These are really very different concepts that are largely independent. They overlap if the selection of physical parameters at the exit of the Planck regime is a quantum process. However, even if physical constants are invariant, we would still need the Many-Worlds type multiverse to make sense of quantum mechanics. Expanding the paper to appropriately treat the Many Worlds interpretation would loose the tight focus that the paper currently has. I suggest, therefore, that the authors clearly state that are focusing on the cosmological multiverse and avoid confusion.

The referee is right that most of our paper focuses on cosmological multiverse scenarios. So we have made a small modification in line 90, and added a footnote in line 93.

The section beginning on Line 95 starts by suggesting “the multiverse encompasses all the multiple possible universes predicted by an underlying theory”. My concern is that this says nothing about those possibilities being realised: the multiverse is proposed simply as a mathematical theorem. I don’t think this is satisfactory starting point, even as a vague one. Surely an essential part of the multiverse concept is that these other universes exist in reality.

We absolutely agree with the referee, and precisely for that reason the sentence continues with “i.e. everything that physically exists, the totality of space and time and its material-energetic content.” However, to avoid any possible confusion we have further emphasized this by writing:“the multiverse encompasses all the multiple possible universes predicted by an underlying theory inosfar as they are actually realized

Line 152. As noted above, The Many Worlds-type of multiverse would still exist. Also, if the inflationary patch in which we live was far larger than our horizon, would we be happy for the causally disconnected regions (which have the same physical parameters) to be referred to as a “multiverse”. I suspect not! The authors should clarify these points.

We have “eliminated” the Many Worlds-type multiverse by inclusion of the footnote mentioned above. Also, we have clarified the sentence by emphasizing: “...and hence without room for a string multiverse, except..”

The discussion of whether a large inflationary patch should be considered as a type of multiverse is a very interesting topic which we (the authors) have in fact discussed several times. So far, we have arrived more or less at what the referee suggests: that if the physical parameters do not vary, this is not really a multiverse, even if the region is larger than our horizon and therefore causally disconnected from a single observer. Indeed, it should be possible to define a set of observers such that there is never any disconnection. However, this description corresponds precisely to Tegmark’s level I multiverse! We have therefore preferred not to include this discussion in the present manuscript but hope to discuss this in future work.

Line 182. Can this entanglement really work? In the situations I’m used to, it is only when the two systems are causally compared that the entanglement becomes apparent. So would the entanglement ever become evident to the observers since they can never share their results. The authors should clarify how an observer might become aware of the entanglement.

We agree with the referee that in order to have an explicit entanglement measure, one needs a classical channel. However, there are several “indirect” issues at play. The existence of quantum correlations among different regions would lead to a different quantum construction (a different Hilbert space), which could have implications on the observables (e.g., the size of the cosmological constant, or some CMB features). Such an effect would therefore not allow to directly pinpoint to the existence of a multiverse, since it could likely be explained by other (single-universe) effects as well. But if a relatively concrete prediction could be made based on a multiverse scenario, and a coincident observation were made, this would surely tremendously increase our confidence in a multiverse scenario.

We would also like to point out that this kind of problem (the impossibility of a direct entanglement detection) is not unique to the multiverse, but occurs in all cases of causally disconnected regions by horizons in General Relativity, such as black holes or cosmological horizons.

We have expanded the discussion in Section 6 to make this clear, see lines 698-704.

Section 4.1

The P(E) term appearing in the equation needs to be defined. This is important, since P(E) is normally taken to the be sum of P(E|H) over all possible H. This is what renders the probability of the truth of H subjective. If the two scientists do not agree on what encompasses “all possible” H, they cannot compute the probability of the truth. This mathematical point underlies much of the following text, and Section 4.2.5. Making this clear would make the text crisper and shorter

We (partially) agree with the referee’s comments, but we are a bit confused by this concrete statement. Also, we are not sure that following his advice would actually make the text crisper and shorter.

First, if the referee means that, contrarily to what has been claimed historically, the essential subjectivity of the Bayesian interpretation can in a certain sense be defended not to lie in P(H), then we agree. This is true in the sense that the P(H) can be divided out to work with Bayesian likelihood factors (as the referee mentions in a further comment) to see which hypothesis is most strengthened by a particular evidence, and then the remaining essence of the subjectivity lies in P(E|H).

However, the key point that we want to stress, as we believe we have written sufficiently explicitly in the paragraph following equation (1), is that the Bayesian model is not well-suited to describe paradigm changes. A paradigm change would imply that somebody who a priori assigned a high P to his point of view, P(H1), and a low P to a competing paradigm P(H2), would change his mind based on some (set of) evidence and correct these absolute probabilities. In that sense, even if a particular piece of evidence would strengthen a competing paradigm more than my own paradigm, i.e.: if the Bayesian likelihood factor for H2 is higher than for H1, this would not convince me to change my paradigm, because I have assigned a much higher prior probability P(H1) than P(H2). This is why we have insisted on the subjectivity of P(H), and avoided defining P(E) explicitly precisely to keep the text as short as possible.

In any case, we have added the explicit definition of P(E) as well as a reference (taking from economics) w.r.t. paradigm change and the limits of Bayesian models. We have also added a brief comment about working with likelihoods (instead of probabilities).

Section 4.2.2

Line 366. The authors need to expand on why there are “serious frictions between cosmological observations and the LCDM model”. I cannot see this to be true. This contentious statement certainly needs to be backed up by recent citations. It is very true that the nature of the dark matter is elusive, and that the model contains puzzling coincidences, such as the similar abundance of matter and dark matter, but these are puzzles rather than an issue with the cosmological model. This is much the same way that the parameters of the Standard Model of particle physics are not explained, but the model itself performs well. This all strengthens the point made, however: observational data do not give any reason to call in the multiverse. The multiverse is only driven by a wish to reach a better understanding of the parameter values.

We absolutely agree with the referee, and thank him for the semantic suggestion: the term “puzzle” is far more adequate here than “friction”. We have modified and expanded this comment, see lines 404-413.

Line 383. This is one of the sections that does not do justice to the Many Worlds interpretation of QM, and its consequent version of the multiverse. It will be better to focus on the cosmological version of the multiverse.

The footnote added previously should make it clear that we are concentrating on cosmological multiverse scenarios and thereby avoid any possible misunderstanding here.

Section 4.2.5

Line 423 This section can be shortened, as noted above.

For the reasons explained above, we are not certain that this would allow us to shorten the text while keeping satisfyingly clear the main point that we want to state. This is that a Bayesian formalism does not change the fact that you need empirical evidence to obtain a paradigm change.

Line 462. Polchinski’s argument is so naive that it cannot possibly be referred to as Bayesian. Since it is erroneous, I would question why it is worth including.

We completely agree with the referee. We have included it because of the references to it in the literature and in popular media, and because it provides what we see as a good opportunity to stress that the lack of explanation for any scientific challenge within a conventional established theory in itself is not a sufficient support for an alternative theory, as we have illustrated with the example from evolution theory.

Line 466. This section is overly negative. While we are unlikely to be able to fully determine the probability of truth, we can focus on the likelihood part of Bayes’ theorem and thus determine whether the parameters of our Universe are likely in a particular Multiverse scenario. We can then compare scenarios on a relative basis. This would interpret the “mediocrity principle” as a familiar likelihood ratio test. While the power of such a test is limited in an n=1 sample, the Bayesian method can still be applied and the implications and caveats quantified. The authors should not dismiss this approach.

We agree with the referee that the Bayesian scheme can be useful to compare different multiverse scenarios, and we have adapted the manuscript correspondingly, see lines 504-510. However, for the reasons mentioned above, as well as the circularity risk that we describe in point (1) of section 4.2.5, we wonder to what extent such methods could be trusted to compare multiverse scenarios with universe scenarios, and that is the point we wanted to stress here. The global statistical assumptions that are used in any application of such cosmic measures are theory-driven (and this theory is itself extrapolated from the n=1 sample), but there is no well-defined mechanism of correcting the theory based on observations, since these are again limited to the n=1 sample which is already implicitly used in the construction of the theoretical background.

There is much discussion about which measure is most adequate, but the point that we have just mentioned seems to us to be generally swept under the carpet in the current literature.

This section would benefit from some discussion of the difficulty of estimating P(E|H) based on anthropic arguments.

A major uncertainty is to decide how similar the Universe needs to be to our own in order that it contains observers that try to measure the cosmological parameters. Dark energy is particularly tricky: is it enough to simply form galaxies? (The weak anthropic principle). Because of recent advances in understanding galaxy formation, this can be calculated reasonably accurately - see Barnes et al., 2018 (MNRAS 477 3727) for an example. A stronger anthropic principle would demand intelligent observers with telescopes capable of mapping the Hubble flow. Some universes will contain only one galaxy per causal patch, so would large telescopes ever be funded? The authors should include some discussion of the fundamental difficulties with such strong anthropic arguments.

We agree with the referee that this is a relevant question. We have not entered into the discussion about the anthropic principle because, although indeed intimately related to the whole multiverse debate, it has already been discussed and dissected so often, and there are so many issues involved here (e.g. the question of galaxy formation, as mentioned by the referee; but also the more philosophical question of how bio-centric, or even anthropo-centric, such arguments should allowed to be), that we do not feel that we could add anything relevant to that discussion without significantly expanding our paper. We therefore prefer to reserve this for future work.

Section 5

The paper quotes Lambda as being 10^120 orders of magnitude smaller than the relevant energy scale. The argument is true, but the origin of the energy could be down to quantum field effects and the matter - dark matter mass asymmetry.

This would put the physics of dark matter (and thus maybe dark energy) with in reach of current accelerators. The energy scale of the dark energy itself is tiny and potentially accessible to desk top experiments. In a few year, the physics of dark matter/ dark energy may well be solved, but this will not easily explain the coincidence of their values. The authors should include discussion of this point.

Following the referee’s suggestion, we have extended the footnote in this section. We appreciate the referee’s optimism, and we agree that dark matter-physics might be within reach of near-future observations. However, we believe that this point of view (especially w.r.t. dark energy) is nevertheless quite speculative. In fact, precisely because of the scale-argument that we explained in this section, we do not really share this optimism w.r.t. dark energy.

Line 568. To finish on a more optimistic note, the forthcoming DESI and EUCLID surveys will measure the growth of large scale structure in the Universe with incredible precision. It is very likely that that these experiments may provide key evidence that drives theorists to a better understanding of (quantum) gravity.

We have reformulated the final part of this section, see lines 632-635. However, as we pointed out in the previous reply, we are not very optimistic that these experiments will provide us with any (reasonably direct) clues w.r.t. quantum gravity, precisely because of the scale problem that we have explained in this section.

This sections asserts that “there is no empirical evidence”. But there is some, albeit of the n=1 variety. The authors should give this more credence. The issue, however, is that Weinberg’s paper sets quite broad limits on the range of Lambda values that could be compatible with our Universe (in the P(E|H) or mediocrity sense). Buosso et al (2007, PhRvD, 76 3513) and more recently Barnes et al 2018 look at this issue in more detail. In particular, in Barnes et al use the latest cosmological simulations to estimate the abundance of galaxies and hence apply a weak anthropic principle to compute P(E|H) for a wide range of Lambda values. They find that our Universe’ value is not at all typical, particularly if the energy argument is used to motivate a log-uniform prior. Lambda values 100x our own still generate abundant stars and galaxies. Disappointingly, the evidence seems to disfavour the Universe (albeit with the n=1 sample caveats). The authors should include discussion of this point.

For the reasons explained in the our previous answers, we have preferred not to enter into a discussion of the anthropic principle. In fact, Weinberg’s prediction is an issue where we tend to agree with Smolin (in his famous correspondence with Susskind) that Weinberg’s prediction has in fact little to do with the anthropic principle per se, and therefore not with the multiverse either. But again, entering into this subject would lead us beyond the scope and length of the paper that we had in mind. Also, although we have not looked at Barnes’ paper in sufficient detail to pinpoint this more exactly, our sensation is that the results obtained there could be strongly model-dependent., and so – even if they are indeed based on state-of-the-art cosmological simulations-- concluding that the evidence seems to disfavour the Universe certainly seems to us to be a rather serious over-extrapolation.

Section 6

Line 604. The paper should acknowledge that n=1 statistics might give some support and discuss what further support would be need to be for the evidence to be conclusive.

See our response to the previous comments, and in particular the brief discussion that we have added in lines 504-510.

Line 609. I am sceptical that the entanglement effects would be apparent to an observer that sees only one universe. The paper should provide a clear example of how this would work without the universes being causally connected in the conventional sense.

See our answer to the point previously raised by the referee and the modifications in lines 698-704.

Round 2

Reviewer 1 Report

I thank the authors for the new and improved version. They have judiciously addressed many of the comments raised. However, there are a few remaining points to address.

2. The multiverse: Nothing new under the suns?

Lines 50-55: the reference to and description of Swedenborg's work is not the right one. Swedenborg first mentions the possibility of other worlds, in the island-Universe sense, in his Principia 1734. In that discussion, he does not specifically propose "other Earths" but rather something like "other celestial spheres" which I interpret as other galactic nebulae or other stellar nebulae. I advise the authors to research this matter to their own satisfaction. 

3. Definition and classification of the multiverse

Line 201: I reiterate the point that the authors might take the opportunity to clarify the notions of theory confirmation, verification and viability, which often cause unnecessary confusion among physicists, but also understand their desire not to overburden the text. At least, the phrase "non-empirical theory confirmation" should be changed to "non-empirical theory assessment" so that the terms are the same in Sections 3 and 4. 

5. Fine-tuning & the multiverse... or is it really a tale of scales?

Line 633: The use of the term "dark structure" appears unsuitable. The term is non-standard and suggests to me a material structure such as the dark matter cosmic web. If that is what is intended, something like "the large-scale distribution of dark matter" might be used instead. If it refers to the properties of unknown dark components such as the dark matter and dark energy, then "dark physics" or "dark sector" seem more appropriate terms. 

Author Response

I thank the authors for the new and improved version. They have judiciously addressed many of the comments raised. However, there are a few remaining points to address.

2. The multiverse: Nothing new under the suns?

Lines 50-55: the reference to and description of Swedenborg's work is not the right one. Swedenborg first mentions the possibility of other worlds, in the island-Universe sense, in his Principia 1734. In that discussion, he does not specifically propose "other Earths" but rather something like "other celestial spheres" which I interpret as other galactic nebulae or other stellar nebulae. I advise the authors to research this matter to their own satisfaction.

We thank the referee again for pointing out this very interesting reference, of which we were not aware. In fact, we must admit that we had some trouble finding it. We have updated the corresponding sentence, see lines 51-55, as well as the reference [13].

3. Definition and classification of the multiverse

Line 201: I reiterate the point that the authors might take the opportunity to clarify the notions of theory confirmation, verification and viability, which often cause unnecessary confusion among physicists, but also understand their desire not to overburden the text. At least, the phrase "non-empirical theory confirmation" should be changed to "non-empirical theory assessment" so that the terms are the same in Sections 3 and 4.

We have changed "non-empirical theory confirmation" into "non-empirical theory assessment", and thank the referee for pointing this out to us.

To be honest, we are not completely sure what the referee means by taking the opportunity to clarify the notions of theory confirmation, verification and viability. Does (s)he mean that we should give explicit definitions? Or point out that these terms are more complicated than what is often assumed naively by physicists? Although we have not addressed these terms explicitly, we hoped that our Section 4 makes clear that the understanding of these terms have evolved throughout the 20th century and are much more complicated than what naive scientific realism might induce physicists to believe.

In any case, we have modified the introduction to Sec.4 by adding lines 228-241 as well as reference [59].

5. Fine-tuning & the multiverse... or is it really a tale of scales?

Line 633: The use of the term "dark structure" appears unsuitable. The term is non-standard and suggests to me a material structure such as the dark matter cosmic web. If that is what is intended, something like "the large-scale distribution of dark matter" might be used instead. If it refers to the properties of unknown dark components such as the dark matter and dark energy, then "dark physics" or "dark sector" seem more appropriate terms.

We agree with the referee, and have replaced "dark structure" by the more commonly used "dark sector".

Reviewer 2 Report

The authors have a made a number of suitable changes in response to my previous report. Following on from those changes, I suggest below a number of further changes that would significantly improve the paper.

Section 4.1

Regarding the definition of P(E), it would be worth noting that this implies that all possible hypotheses (models) have been considered. 

In practice, only a limited range of hypotheses are summed over, and P(E) becomes a somewhat subjective assessment.

The likelihood ratios also have the advantage of factoring P(E) out of the comparison, but, as the authors note, this only really works to compare two alternatives within the same paradigm. I do not see why P(E|H) should be subjective: the hypothesis should contain all the conjectures necessary for the calculation.

However, if we view a “paradigm shift” as a move to considering a new range of hypotheses, H2, the subjectivity becomes evident. Assuming P(E|H2) is much larger, the new hypothesises lead to a reassessment of P(E), and thus to a down-weighting down-weighting of P(H1) even though the calculation of (PE|H1) has not changed.  The new hypothesis is simply a better explanation of the data.

Of course, unless the evidence is particularly strong, the Prior factors assigned by different scientists will lead to an unclear outcome, but the beauty of the Bayesian approach is that the subjectivity has been made explicit.

This interpretation of Bayes’ theorem seems consistent with the “hypothesis testing” paradigm proposed in the reference rather than sitting outside of Bayes approach.

I agree that the authors do not want to over complicate the text addressing this point, adding a sentence on the role of P(E) would be sufficient.

Section 4.2.2 It would be helpful to give some references here. At present the comment “perhaps even major revisions” does not seem justified.

Line 508. (Original line 466).

“There also exists a challenge of understanding how such a scenario based on an n=1 construction could help us deciding whether a multiverse scenario is really needed, rather than a single universe”

I am still puzzled by the authors reluctance to adopt a Bayesian treatment. We can include the “single” universe not the range of hypotheses by considering a “multiverse” through-out which all the physical constants have the same values (and hence the same lambda etc) (ie., this “multiverse” is just a universe that is larger then our horizon). We can then form the likelihood ratio: our “constant” universe will have a likelihood of 1 (by construction), but the “variable” multiverse will have some other value. Although this ratio can’t tell us to reject the “constant” multiverse, it can tell us if the “variable” scenario is viable, or whether supporters should be uncomfortable.

Although the authors want to reserve a discussion of the role of anthropic arguments to a future paper, it surely deserves a statement in the text saying this, and giving some pointers to the literature.

Line 632 (original line 568). Please add references.

Footnote, Section 5. It would be worth noting that kc=E_weak still makes the energy too large.

Section 7.  

The paper should include some conclusion regarding the use of n=1 evidence. After all, it is all we are ever likely to have. I am still unclear what the authors have in mind when they say “to discriminate emergent ideas and to look for the possible testability of different cosmological scenarios involving either a single universe or a multiverse in any of the various multiverse definitions.” The paper would benefit from a clear recommendation.

Author Response

The authors have a made a number of suitable changes in response to my previous report. Following on from those changes, I suggest below a number of further changes that would significantly improve the paper.

Section 4.1

Regarding the definition of P(E), it would be worth noting that this implies that all possible hypotheses (models) have been considered.

In practice, only a limited range of hypotheses are summed over, and P(E) becomes a somewhat subjective assessment.

The likelihood ratios also have the advantage of factoring P(E) out of the comparison, but, as the authors note, this only really works to compare two alternatives within the same paradigm. I do not see why P(E|H) should be subjective: the hypothesis should contain all the conjectures necessary for the calculation.

However, if we view a paradigm shift as a move to considering a new range of hypotheses, H2, the subjectivity becomes evident. Assuming P(E|H2) is much larger, the new hypothesises lead to a reassessment of P(E), and thus to a down-weighting down-weighting of P(H1) even though the calculation of (PE|H1) has not changed. The new hypothesis is simply a better explanation of the data.

Of course, unless the evidence is particularly strong, the Prior factors assigned by different scientists will lead to an unclear outcome, but the beauty of the Bayesian approach is that the subjectivity has been made explicit.

This interpretation of Bayes theorem seems consistent with the hypothesis testing paradigm proposed in the reference rather than sitting outside of Bayes approach.

I agree that the authors do not want to over complicate the text addressing this point, adding a sentence on the role of P(E) would be sufficient.

Following the indications of the referee, we have modified lines 349-350.

Section 4.2.2 It would be helpful to give some references here. At present the comment perhaps even major revision does not seem justified.

We have added the two recent references [80,81].

Line 508. (Original line 466).

There also exists a challenge of understanding how such a scenario based on an n=1 construction could help us deciding whether a multiverse scenario is really needed, rather than a single universe

I am still puzzled by the authors reluctance to adopt a Bayesian treatment. We can include the single universe not the range of hypotheses by considering a multiverse through-out which all the physical constants have the same values (and hence the same lambda etc) (ie., this multiverse is just a universe that is larger then our horizon). We can then form the likelihood ratio: our constant universe will have a likelihood of 1 (by construction), but the variable multiverse will have some other value. Although this ratio can tell us to reject the constant multiverse, it can tell us if the variable scenario is viable, or whether supporters should be uncomfortable.

We are not opposed to a Bayesian treatment per se. In our discussion of the Bayesian treatment, we just wanted to insist that –contrarily to the impression one might obtain when reading contemporary literature on multiverse, quantum gravity and final theories-- Bayesian approaches are far from the miracle solution that will allow us to decide whether there really is a multiverse or not (or whether string theory is viable or not; or...). In this context, we wanted to stress that the alleged paradigm change from the universe to a multiverse is only in an embryonic state, and there is no clear evidence of the necessity for such a change as yet; certainly not at the empirical level, and (as we have argued in Sec. 5) probably not (yet) at the theoretical level either.

The questions raised by the referee, about the viability of a multiverse approach, although related, is quite different. In our opinion, the brief answer is the following. Since we know the universe exists, any statistical distribution based on theories (such as string theory or inflation) that supposedly describe the known physics of our universe should be viable by construction. Otherwise, something went very wrong in constructing these theories.

We understand that the referee’s suggestion goes beyond this straightforward answer and proposes to quantify this viability through a Bayesian treatment. But although the referee’s suggestion certainly sounds interesting, this would need to be worked out in detail, in order to escape Ellis’ criticism that “the statistical argument only applies if a multiverse exists; it is simply inapplicable if there is no multiverse: we cannot apply a probability argument if there is no multiverse to apply the concept of probability to.''

Although the authors want to reserve a discussion of the role of anthropic arguments to a future paper, it surely deserves a statement in the text saying this, and giving some pointers to the literature.

Following the recommendation of the referee, we have added a footnote on page 3.

Line 632 (original line 568). Please add references.

We have added footnote 6 and the two references [102,103].

Footnote, Section 5. It would be worth noting that kc=E_weak still makes the energy too large.

We have modified the footnote to incorporate the referee’s suggestion.

Section 7.

The paper should include some conclusion regarding the use of n=1 evidence. After all, it is all we are ever likely to have. I am still unclear what the authors have in mind when they say to discriminate emergent ideas and to look for the possible testability of different cosmological scenarios involving either a single universe or a multiverse in any of the various multiverse definitions. The paper would benefit from a clear recommendation.

Much as we would like to give a clear recommendation, we believe that our current recommendation is about the most concrete that is possible at this point: to keep in mind that scientific progress requires an interplay between theory-building and empirical assessment (as discussed at length throughout the manuscript and stressed again in the first paragraph of the conclusions), and that therefore frameworks similar to the one we discussed at some length in Section 6 deserve more attention (as summarized just before the fragment cited by the referee).